# Cobalt Ferrite (CoFe₂O₄) Spinel as a New Efficient Magnetic Heterogeneous Fenton-like Catalyst for Wastewater Treatment

**Maria Alice Prado Cechinel** [1] 📷, **João Lucas Nicolini** [2], **Pedro Monteiro Tápia** [2],
**Edgar Andrés Chavarriaga Miranda** [3], **Sarah Eller** [4], **Tiago Franco de Oliveira** [4] 📷, **Fabiano Raupp-Pereira** [2] 📷,
**Oscar Rubem Klegues Montedo** [2] 📷, **Tiago Bender Wermuth** [2] and **Sabrina Arcaro** [2,*] 📷

[1] Department of Chemical and Food Engineering (EQA), Federal University of Santa Catarina (UFSC), Florianópolis 88040-900, Brazil; cechinel.maria@gmail.com
[2] Biomaterials and Nanostructured Materials Group, Technical Ceramics Laboratory (CerTec), Postgraduate Program in Materials Science and Engineering, The University of Southern Santa Catarina (UNESC), Av. Universitária 1105, Criciuma 88806-000, Brazil; joaolucasnicolini@gmail.com (J.L.N.); pedromt01@unesc.net (P.M.T.); fraupp.pereira@gmail.com (F.R.-P.); okm@unesc.net (O.R.K.M.); tiago.haine@gmail.com (T.B.W.)
[3] Department of Basic Sciences, Universidad Católica Luis Amigó, Transversal 51 A # 67B-90, Medellín 050034, Colombia; eachavar@unal.edu.co
[4] Pharmacosciences Department, Federal University of Health Sciences of Porto Alegre, Porto Alegre 90050-170, Brazil; sarahcarobini@hotmail.com (S.E.); oliveira@ufcspa.edu.br (T.F.d.O.)
* Correspondence: sarcaro@unesc.net

**Abstract:** For the first time, cobalt ferrite spinel (CoFe₂O₄) was used as a catalyst in the Fenton process for Remazol Red RR dye degradation in water. CoFe₂O₄ was synthesized via gel combustion using tris(hydroxymethyl)aminomethane as an alternative fuel in one step with a ratio of Ψ = 0.8. Its structural, surface optics, magnetic properties, and the optimal conditions of the Fenton reagents for dye degradation were evaluated. The saturation magnetization and remanence (Ms and Mr, respectively) for the as-prepared powder were 65.7 emu/g and 30.4 emu/g, respectively, and the coercivity (Hc) was 1243 Oe, indicating its ferromagnetic nature and suitability as a magnetic catalyst. Red Remazol RR dye degradation tests were performed using the Fenton process to evaluate the influence of the catalyst dosage and H₂O₂ concentration. The tests were performed in a batch reactor in the dark with constant agitation for 24 h. The best result was obtained using 1 g/L of catalyst with a dye degradation of 80.6%. The optimal concentration of H₂O₂ (1.0 M) resulted in 96.5% dye degradation. Nanoparticle recyclability testing indicated that the material could be satisfactorily reused as a catalyst for at least three cycles. The potential use of the CoFe₂O₄ synthesized in this study as a catalyst for dye degradation by the Fenton process was demonstrated.

**Keywords:** spinel ferrites; cobalt ferrite; advanced oxidative process; textile dye; reuse; degradation pathway

## 1. Introduction

The increasing concentration of organic contaminants in groundwater and surface water has become a source of great concern. The textile industry is a major consumer of synthetic dyes and water. Therefore, it is one of the primary industries responsible for generating and discharging liquid effluents. The discharge of pollutants from the textile industry into aquatic environments can cause serious health and environmental problems, as well as negative visual impacts due to water coloration [1–3].

Many treatment processes are used to remove organic contaminants from contaminated water in the textile industry. Conventional treatment processes, which include chemical, physical, and biological treatment steps [4], do not always provide satisfactory results because many organic substances produced by the chemical industry are toxic or

resistant to biological treatment [5–7]. For example, textile industry effluents have a low level of biological degradation, which makes conventional treatment processes even more challenging [8]. An effective alternative for the degradation of these organic pollutants is the use of oxidation processes such as the Fenton reaction, in which iron ions interact with hydrogen peroxide in an acidic medium to form highly reactive hydroxyl radicals ($\cdot$OH) [9]. These radicals react non-selectively with almost all organic compounds, oxidizing them to intermediates, such as alcohols, carboxylic acids, or aldehydes, and then to water and carbon dioxide [10].

The Fenton reaction can occur in the homogeneous form, where iron ions and $H_2O_2$ react directly in solution to produce $Fe^{3+}$, OH, and $HO^-$, or in the heterogeneous form, where a solid iron-based catalyst is used to initiate the decomposition of $H_2O_2$ into hydroxyl radicals [11]. Metallic spinel ferrites with the general formula $MFe_2O_4$ ($M = Fe^{2+}$, $Zn^{2+}$, $Mn^{2+}$, $Cu^{2+}$ or $Co^{2+}$) have attracted great interest from the scientific community because of their excellent physicochemical stability and magnetic recovery performance [12]. Among the different types of ferrites, cobalt ferrite ($CoFe_2O_4$) has a spinel structure and has been shown to be an excellent material for environmental remediation owing to its high catalytic activity, stable crystalline structure, low metal dissolution, high saturation magnetization, large specific surface area, and ability to easily separate water magnetically [13–16]. These properties distinguish it from other heterogeneous iron precursor catalysts used in Fenton processes [17–19]. Cobalt ferrite has also been used as a solid catalyst for ozonation reactions [16,20]. Because it is a low-bandgap semiconductor (approximately 1.08 eV) and is highly stable as a photocatalyst [21,22], $CoFe_2O_4$ can also be a more advantageous alternative to $TiO_2$ and other photoactive materials, such as the perovskite structure ($ABO_3$), in photocatalytic processes [23–26].

The various technological applications of $CoFe_2O_4$ have led to an extensive search for methods to source it. Some of these techniques include the sol–gel [27], coprecipitation [28], gel combustion [29], and hydrothermal [15] methods. Combustion synthesis has advantages over other techniques, such as the ability to synthesize nanometric powders with the desired crystalline structure in a single step. This eliminates the need for post-combustion heat treatment, making the process environmentally attractive owing to its low energy and time consumption [30]. This method is often used to obtain complex oxides and involves an exothermic and self-sustaining reaction between a fuel and an oxidant, allowing high temperatures to be reached in short reaction times with a simple experimental setup [31]. Previously [29], this research group synthesized new nanometric $CoFe_2O_4$ with interesting physicochemical properties for application in the environmental field as a catalyst in the Fenton process.

For the first time, in this study, $CoFe_2O_4$ nanoparticles were synthesized using the combustion method as catalysts in a Fenton-like reaction for the degradation of textile dyes present in water. Our results indicate that a simple system containing $CoFe_2O_4$ at low concentrations can result in high degradation rates of Remazol Red RR dye. In addition, analysis of the degradation byproducts showed significant mineralization into simple organic acids, demonstrating the remarkable effectiveness of this catalytic system for treating water contaminated with persistent pollutants.

## 2. Materials and Methods

### 2.1. Synthesis and Characterization of the Cobalt Ferrite (CoF$_{e2}$O$_4$) Spinel Catalyst

$CoFe_2O_4$ was synthesized via gel combustion using tris (hydroxymethyl) aminomethane (TRIS, $C_4H_{11}NO_3$, Neon, 99%) as an alternative fuel. Iron nitrate nonahydrate ($Fe(NO_3)_3.9H_2O$, Neon, 98%) and cobalt nitrate hexahydrate ($Co(NO_3)_2$-$6H_2O$, Sinth, 98%) were used as oxidizers.

The synthesis of 2 g of spinel ferrite with a composition of $CoFe_2O_4$ and $\Psi = 0.8$ was performed using 6.890 g of $Co(NO_3)_2.6H_2O$ and 2.5098 g of $Fe(NO_3)_3.9H_2O$ dissolved in 50 mL of distilled water. Subsequently, 2.4899 g of fuel (TRIS) was added under magnetic stirring (HSC-F20500101, Velp, Usmate Velate, Province of Monza and Brianza, Italy) at 200 rpm for 20 min. The solution was slowly evaporated at 70 °C until a brown gel was

formed. Finally, the gel was heated with a flame for 30 s until the combustion reaction occurred. A detailed description of the synthesis parameters can be found elsewhere [29].

The crystalline phases were determined by X-ray diffraction (XRD) on Philips equipment (Amsterdam, The Netherlands) using Cu-K $\alpha$ radiation ($\lambda$ = 1.54184 Å) at 40 kV and 40 mA with a 2$\theta$ range of 10–70°. The samples were analyzed at 0.02°/2s. The XRD patterns were compared with the Inorganic Crystal Structure Database (ICSD) using the X'Pert HighScorePlus® software to identify the crystalline phases. The crystallite sizes were determined after refining the structures using the Rietveld method [32]. The network parameters, occupancy, third-degree polynomial, peak scale, and shape factors were refined. The quality of fit (GoF) was used to describe the refinement quality. The lower limit of the average crystallite size was calculated using the Scherrer equation [32].

Raman spectra of the prepared powders were obtained under 532 nm laser excitation (Renishaw no Via Spectrometer System). The microstructures of the prepared powders were analyzed by field-emission scanning electron microscopy (FE-SEM; model S-4100, Hitachi Ltd., Tokyo, Japan) at 20 kV and transmission electron microscopy (TEM) at an accelerating voltage of 100 kV (JEM-1011 TEM, JEOL USA, Inc., Peabody, MA). The samples were prepared on a metal stub using an adhesive and coated with gold and palladium. The zeta potentials of the samples were measured using a NanoZ instrument (Litesizer 5000, Anton Paar GmbH, Graz, Austrian). The measurements were performed by diluting 0.1 g of each mixture in 1000 mL of distilled water using KCl ($10^{-2}$ M), HCl ($10^{-1}$ M), and KOH ($10^{-1}$ M) as inert electrolytes. The zeta potential ($\xi$) was calculated automatically from the electrophoretic mobility of the samples.

The light absorption curves of the synthesized ferrites were obtained using diffuse reflectance spectroscopy (DRS, Cary 5000, Agilent equipment, Santa Clara, California, USA) with an integrating sphere (DRA 1800). The gap energy was determined using the Kubelka–Munk function [33]. The magnetic properties of the powders were measured at room temperature (298 K) using a vibrating-sample magnetometer (Model EZ9, Microsense, Lowell, Massachusetts, USA). The Mossbauer spectra of the powder samples were obtained using a Wissel instrument (Wissenschaftliche Elektronik GmbH, Würmstraße, Starnberg, Germany) operating in the constant acceleration mode with a Co57 (Rh matrix) source. The spectra collected in the transmission geometry and at room temperature were analyzed using the least-squares method for the discrete Lorentzian line at each hyperfine site (jbat). The deviation of the isomer was related to metallic iron.

### 2.2. Remazol RR Dye Degradation

The RR dye (chemical formula: $C_{19}H_{16}N_2Na_2O_{11}S_3$) used in this study as a contaminant was supplied by a partner textile industry. It is a monoazo-type dye with vinylsulfonyl (VS) and monohalogentriazine (MHT) reactive groups. The azo group acts as a chromophore (Figure 1) [34,35]. The solution dye content was measured in all the catalytic tests using UV-Vis spectrophotometry (Shimadzu UV-1800, Kyoto, Japan) at the maximum absorption wavelength ($\lambda_{max}$ = 525 nm). An analytical calibration curve, Abs = 0.0228 $C_{dye}$ + 0.0035, with $R^2$ = 0.9992 was constructed using dye concentrations ranging from 0.5 to 40 mg/L. The curve showed a dye detection limit of 0.585 mg/L and a quantification limit of 1.949 mg/L.

To evaluate the influence of the cobalt nanoferrite ($CoFe_2O_4$) dose on the degradation of RR dye using a Fenton-like process, 50 mL of 40 mg/L RR dye solution and 100 mM $H_2O_2$ were placed in contact with catalyst mass/volume ratios ranging from 0.2 to 4.0 g/L in 100 mL Erlenmeyer flasks. The reaction flasks were covered to prevent the influence of light on the results. The assay was performed in duplicate. The system was kept at room temperature (24 $\pm$ 2 °C) under constant orbital stirring (100 rpm) (Fisatom 713D, São Paulo, Brazil) for 24 h. Then, the samples were filtered through 0.45 µm cellulose acetate membrane filters, and the pH was measured using a pH meter (PH0-14 Kasvi, São José dos Pinhais, Paraná, Brazil). The residual dye concentration after the reaction was measured using UV-Vis spectroscopy (Shimadzu UV-1800, Kyoto, Japan).

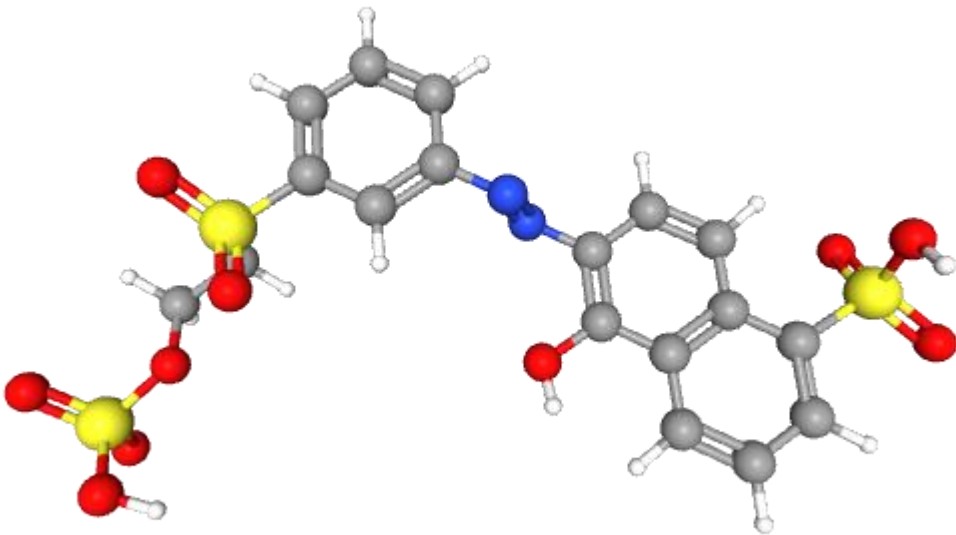

**Figure 1.** Molecular 3D structure of Remazol Red RR dye [36].

Once the optimum $CoFe_2O_4$ dosage was determined, the optimum concentration of $H_2O_2$ for the effective degradation of the RR dye by a Fenton-like process was evaluated; 50 mL of a 40 mg/L RR dye solution with $H_2O_2$ concentrations ranging from 20 to 200 mM was placed in contact with the catalyst at a previously defined optimum mass/volume ratio. The stirring and temperature conditions were the same as those used in previous tests. The samples were then filtered, their pH was evaluated, and the residual dye concentration was measured using UV-Vis spectroscopy.

After defining the optimal conditions for the $CoFe_2O_4$ and $H_2O_2$ dosages, kinetic analysis of the RR dye degradation by a Fenton-like process was carried out for a solution with a dye concentration of 40 mg/L. The test was performed in a 2 L beaker; 1 L of the RR dye and $H_2O_2$ solution and the $CoFe_2O_4$ catalyst in the optimal proportions, as previously defined, were maintained under constant mechanical agitation (ARE, Velp Scientifica, Betim, Minas Gerais, Brazil) at 120 rpm and at room temperature (19 ± 1 °C). Every 15 min, an aliquot of the solution was collected, and the dye concentration was evaluated using UV-Vis spectroscopy. Pseudo-first- and pseudo-second-order kinetic models were used to fit the experimental data and obtain the intrinsic constants of the kinetic degradation rates. Further information regarding these models can be found elsewhere [37,38].

*2.3. Catalyst Reusability*

Catalyst reuse is a fundamental aspect of heterogeneous Fenton systems, especially for large-scale practical applications. Reuse tests are critical for evaluating the effectiveness of a catalyst in successive cycles of use to demonstrate that the material has a longer useful life and to avoid frequent disposal, increasing its economic viability. In the $CoFe_2O_4$ reuse tests, 50 mL of the problem solution containing 40 mg RR/L and 200 mM $H_2O_2$ was added to a 250 mL Erlenmeyer flask containing a 1.0 g/L catalyst. The solution was kept under orbital stirring at room temperature (20 ± 1 °C) for 24 h, which was followed by decanting of the solid phase for 30 min. The liquid phase was then removed from the flask and analyzed using UV-Vis spectroscopy to determine the concentration of the dye. To complete the reuse cycle, the flask containing the solid phase was placed in an oven at 60 °C overnight to remove residual moisture, and the mass of $CoFe_2O_4$ present in the flask was determined using an analytical balance. A new cycle was initiated by adding the same amount of 40 mg RR/L solution and 200 mM $H_2O_2$ to the flask containing the catalyst and keeping it under orbital agitation for another 24 h. Three catalyst cycles were performed to degrade the RR dye.

*2.4. RR Dye Degradation Pathway*

Time-dependent identification of the Remazol dye degradation pathway was performed using an LCMS-8045 triple quadrupole mass spectrometer (Shimadzu, Kyoto, Japan). Mass spectra were obtained under the following conditions: electrospray ionization mode, negative ([M-H]-); desolvation line temperature, 250 °C; heating block temperature, 400 °C; heating gas, 10 L/min; drying gas, 10 L/min; and nebulizing gas, 3 L/min. Data were collected in the SCAN mode within an m/z range of 50–900. The analysis was conducted by directly infusing aqueous samples into the mass spectrometer system using a syringe pump at a 3 μL/min flow rate. The generated data were processed using the LabSolutions software 5.6 (Shimadzu, Kyoto, Japan).

## 3. Results and Discussion

### 3.1. Characterization of $CoFe_2O_4$ Nanoparticles

Table 1 shows the influence of different fuels on the synthesis of the $CoFe_2O_4$ nanoparticles using combustion synthesis. The synthesis of cobalt ferrite is a crucial process in various technological applications, and the choice of fuel and synthesis temperature can have a significant impact on the efficiency. In this analysis, the advantage of using Tris (with an oxidant/fuel ratio of $\Psi = 0.8$) as a fuel in cobalt ferrite synthesis at 140 °C to other fuels reported in the literature was compared in terms of energy savings. Synthesis at 800 °C with citric acid has high energy consumption and requires high-temperature equipment. In contrast, synthesis at 500 °C with tartaric, citric, and oxalic organic acids may be more energy-efficient compared to that at 800 °C but can still be relatively energy-intensive because of the required temperature. Moreover, synthesis at 700 °C using glycine or citric acid can also be energy-intensive because of the high temperature required.

**Table 1.** Different fuels used in combustion synthesis of $CoFe_2O_4$.

| Fuel | Temperature (°C) | Reference |
|---|---|---|
| Citric acid | 800 | [39] |
| Tartaric acid, citric acid, and oxalic acid | 500 | [40] |
| Glycine or citric acid | 700 | [41] |
| Urea | 700 | [42] |
| 6-aminohexanoic acid | 230 | [43] |
| Caffeine and citrulline | 200–250 | [44] |
| Tris with $\Psi = 0.8$ | 140 | [29] |

Although glycine and citric acid are common fuels used in synthesis, the temperature remains a challenge in terms of energy consumption. Synthesis at 700 °C with urea also requires a significant amount of energy. In contrast, synthesis at 230 °C with 6-aminohexanoic acid indicates a lower energy consumption. Conversely, synthesis in the relatively low 200–250 °C range with caffeine is more energy-efficient compared to those using higher temperatures. Finally, synthesis at 140 °C with Tris and an oxidant/fuel ratio of $\Psi = 0.8$ shows a clear advantage in terms of energy savings. A lower temperature reduces the energy consumption, and an optimal fuel–oxidant ratio contributes to the process efficiency. This choice may be more sustainable and economically viable for industrial applications that seek to minimize energy consumption.

To confirm the formation of the cobalt ferrite phase, Figure 2 shows the X-ray diffractogram (Figure 2a) and Raman spectra (Figure 2b). In Figure 2a, only the reflections related to $CoFe_2O_4$ (ICSD 1533163, special group Fd3m) are shown. The network parameter calculated using the Rietveld refinement was 8.3850 Å (Table 2).

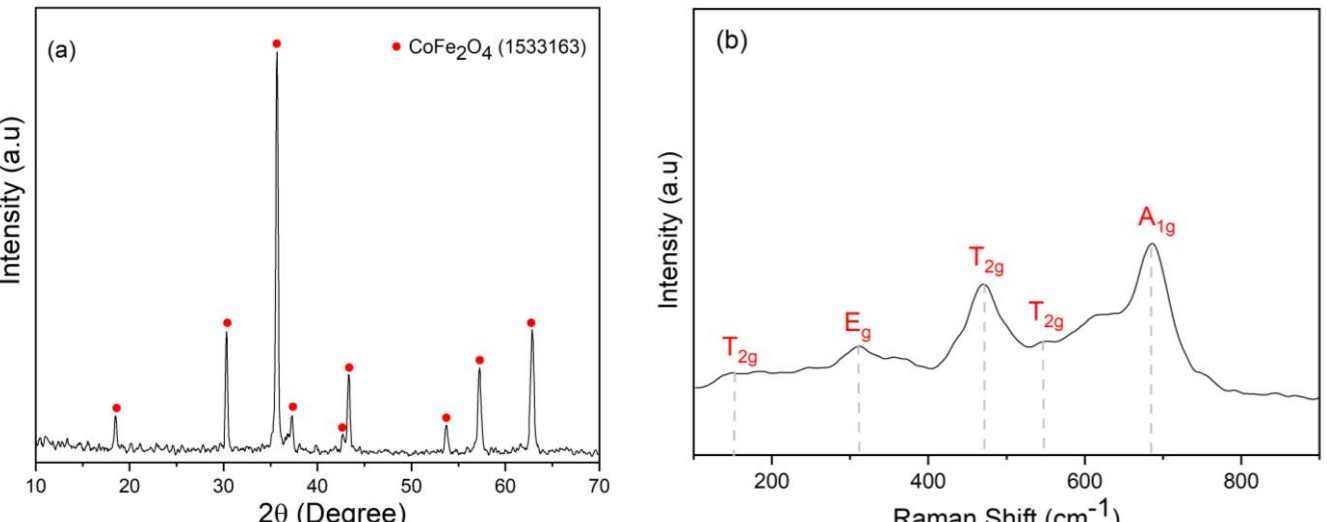

**Figure 2.** XRD (**a**) and Raman (**b**) patterns of the CoFe$_2$O$_4$ sample synthesized by the combustion method.

**Table 2.** Lattice parameter, crystallite size, and surface area of the synthesized CoFe$_2$O$_4$.

| Structural and Morphological Properties | |
|---|---|
| Lattice parameter (Å) | 8.3850 |
| Crystallite size (nm) | 35 |
| Surface area (m$^2$/g) | 59 |
| Magnetic properties | |
| Remnant magnetization (Mr) (emu/g) | 30.4 |
| Coercivity (Hc) (Oe) | 1243 |
| Quadrature (S–Mr/Ms) | 0.46 |
| Saturation magnetization (Ms) (emu/g) | 65.7 |

These results demonstrate that the ferrite synthesis was successful and similar to the results obtained in a previous study [29]. The crystallite sizes were determined from the pronounced broadening of the diffractogram reflections, which revealed crystallites in the nanometric range (Table 1). The crystallite size was 35 nm. Figure 2b confirms that the sample synthesized using the combustion method had all the vibrational modes characteristic of CoFe$_2$O$_4$ and no secondary-phase modes.

The Raman spectra revealed the five active Raman modes, such as T$_{2g}$ (3), E$_g$ (1) and A$_{1g}$ (1), at 151, 311, 470, 547, and 685 cm$^{-1}$, respectively [45,46]. In the reverse spinel, the tetrahedral sites of the ferrite were occupied by half of the Fe$^{3+}$ cations, whereas the other Fe$^{3+}$ and Co$^{2+}$ cations were distributed in the octahedral sites. The bands obtained at 685 cm$^{-1}$ were attributed to A1g symmetry, which has symmetrical stretching with the oxygen atom bound to metal ions (MO) located in tetrahedral sites [47]. The two modes obtained in the band at 470 and 547 cm$^{-1}$ were related to antisymmetric stretching and bending of the metal–oxygen bond. The E$_g$ mode located at 310 cm$^{-1}$ corresponded to symmetrical bending of the metal–oxygen bond. The band located at 150 cm$^{-1}$ corresponded to the translational movement of metal ions in the tetrahedral site. The vibrational modes observed at low frequencies (~150 cm$^{-1}$ to 550 cm$^{-1}$) were attributed to the symmetry of oxygen contact with metal ions located in octahedral sites, and this can be called BO$_6$. These modes represent the symmetric and antisymmetric bending of the oxygen atom with metal ions (MO) in the octahedral sites [45].

Figure 3 shows the SEM (Figure 3a) and TEM (Figure 3b) images of the cobalt ferrite (CoFe$_2$O$_4$) synthesized by the combustion method.

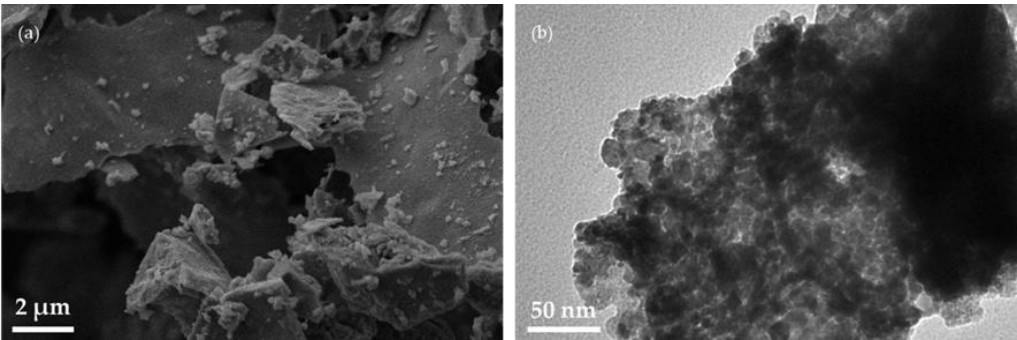

**Figure 3.** SEM (**a**) and TEM (**b**) images of the $CoFe_2O_4$ powders synthesized by the combustion method.

A sample prepared using $\Phi = 0.8$ was analyzed by SEM. Figure 3a shows the microstructure composed of large agglomerates, which was similar to previous studies [29,44]. Because of the synthesis method used, the $CoFe_2O_4$ particles were highly reactive, and it is considered that these agglomerates were composed of nanoparticles. Thus, TEM could be used to visualize their morphology at the nanoscale level. The images obtained are shown in Figure 3b. The images revealed crystallite sizes of approximately 20 and 40 nm for the cobalt ferrite nanoparticles. The spherical shape and high degree of aggregation obtained was as expected for a magnetic material.

Table 1 shows the specific surface area of this sample, which was 59 $m^2/g$. The small crystallite size (35 nm) and the large specific surface area obtained by BET are extremely interesting for applications involving photo-Fenton processes. One of the advantages of using ferrite nanoparticles in water treatment processes is their magnetic properties, which facilitate the separation of the fluid phase from the particulate phase. Table 1 also shows the magnetic properties (saturation magnetization, Ms, remnant magnetization, Mr, coercivity, Hc, and quadrature (S)) based on Figure 2a, which shows the magnetic behavior observed using the vibrating sample magnetometer (VSM).

Figure 4 shows the hysteresis loop (a) and the Mössbauer spectrum (b) measured at room temperature for the $CoFe_2O_4$ synthesized using the combustion method. The saturation magnetization (Ms) and remanence (Mr) were 65.7 emu/g and 30.4 emu/g, respectively, and the coercivity (Hc) was 1243 Oe. These results were consistent with those reported in the literature, where Mr and Ms were 36 emu/g and 60.5 emu/g, respectively, and Hc was 1305 Oe [48]. The magnetic properties of spinel nanoparticles are highly influenced by the distribution of cations in tetrahedral (site A) and octahedral (site B) locations. Due to the AB superexchange interaction (the interaction between the $Fe^{3+}$ cations of site A and the $Fe^{3+}$ of site B from a nonmagnetic oxygen anion), the AA and BB exchange interactions dominate. Thus, the cation distribution plays a significant role in determining the magnetic properties of this type of material. Finally, the wide characteristic hysteresis loops of the $CoFe_2O_4$ indicate a hard magnetic character [49].

Mössbauer spectroscopy provides profound insights into the precise atomic environment encompassing the iron cations within the framework of a spinel lattice. The Mössbauer spectra depicted in Figure 4b illustrate the distinctive sextet patterns that characterized the $CoFe_2O_4$ sample; within the Mössbauer spectrum, a hexaline hyperfine pattern emerged, unequivocally signifying the existence of two sextets attributed to the arrangement of $Fe^{3+}$ ions at the tetrahedral (A) and octahedral (B) positions. Significantly, isomer shift values within the A-sites were consistently more modest than those within the B-sites, a contrast attributed to the relatively larger $Fe^{3+}$-$O_2^-$ bond distances present in the B-sites, resulting in a subdued covalency effect. This diminished isomer shift value correlated with an augmented concentration of s-electrons within the Fe atoms at the A-sites, which originated from ligand-to-metal charge transfer ($\sigma$L-M) interactions. This phenomenon is similar to the observations made in analogous contexts [17,50,51].

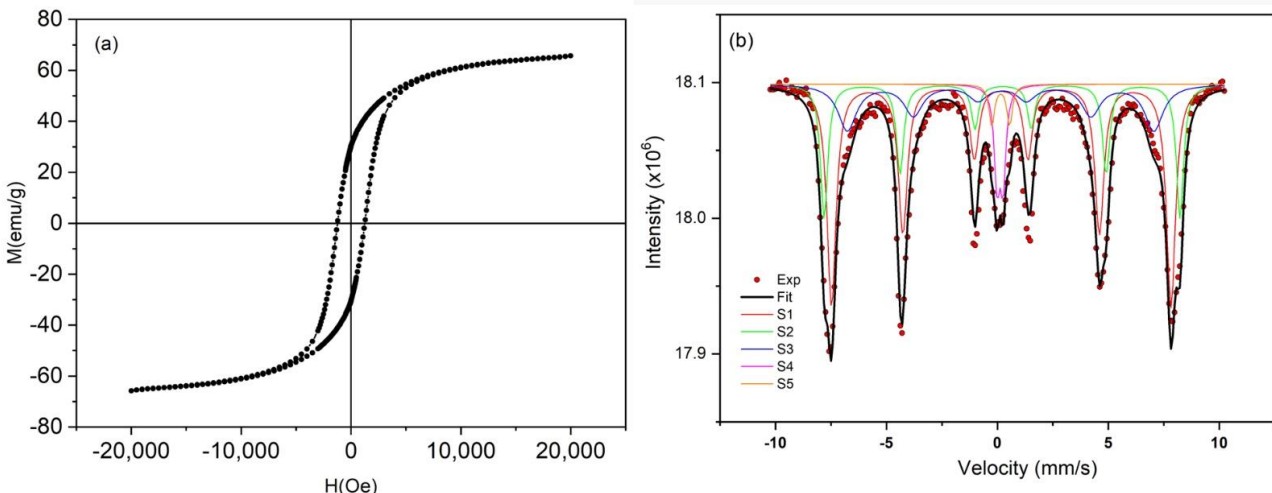

**Figure 4.** (**a**) M-H curves of CoFe$_2$O$_4$ powder at room temperature and (**b**) Mössbauer spectrum of the sample synthesized by the combustion method.

To confirm the potential application of the synthesized CoFe$_2$O$_4$ ferrite in the Fenton process, its optical properties were analyzed using the diffuse reflectance spectra obtained by UV-Vis spectrophotometry. The bandgap energies (Eg) of the samples (Figure 5) were calculated using the traditional semiconductor method [33], corresponding to the transition of electrons from the valence band to the conduction band. The calculated optical transitions were considered indirect, considering the coupling mechanism with the phonons of the crystal structure at room temperature and disregarding other effects owing to the nature of the transition. Thus, the results obtained using the Kubelka–Munk method indicated a bandgap value of 1.16 eV.

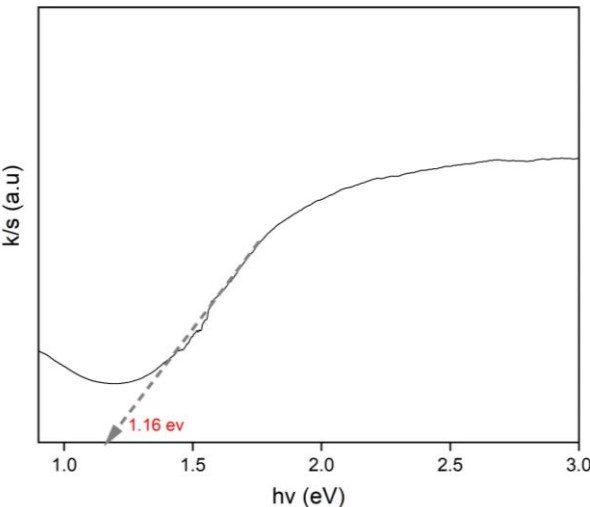

**Figure 5.** Bandgap of the CoFe$_2$O$_4$ powders synthesized by the combustion method.

The results obtained in this study were remarkably close to those previously reported in the literature. For example, Badizi et al. [52] obtained energy values in the range of 1.10 eV. The optical gap in the pure cobalt ferrite nanoparticles was caused by the transformation of the metallic charge. In fact, a charge-transfer transition occurred between the Fe$^{3+}$ and Fe$^{2+}$ in the crystal lattice. The presence of Fe$^{3+}$ and Co$^{2+}$ could create a conduction band level, and the electron could be promoted from the valence band to this level. That is, variations in the degree of ferrite inversion could completely change the optical gap. This result also shows that the gap obtained was relatively small when compared to UV-Vis-activated

catalysts such as $TiO_2$ (3.2 eV) or magnetite (<3 eV), indicating excellent potential for applications where low bandgaps are required, such as in the degradation of organic compounds by photo-Fenton or photocatalytic processes [53].

The dispersion of nanoparticles in liquid media is strongly affected by the pH at which the nanoparticles are found, as the pH directly influences the surface charge of the particles. The zeta potential is related to the surface charge present on the nanoparticles; a large/small value of the zeta potential indicates a greater/minor electrostatic repulsion between the nanoparticles. For magnetic nanoparticles, this electrostatic repulsion opposes the magnetic attraction between the nanoparticles. Figure 6 shows the zeta potential curves as a function of the solution pH for the $CoFe_2O_4$ powders obtained by combustion synthesis. Initially, the point of zero charge (PZC) occurred at pH 4.8. At this pH, there was little electrostatic repulsion to prevent the particles from coming together, i.e., at this point, the magnetic attraction exceeded the electrostatic repulsion, which led to a greater agglomeration of the nanoparticles. Moreover, the nanoparticles tended towards colloidal stability at pHs above 10, where the zeta potential was approximately −20 mV, or at pHs below 1, where the zeta potential was approximately 20 mV.

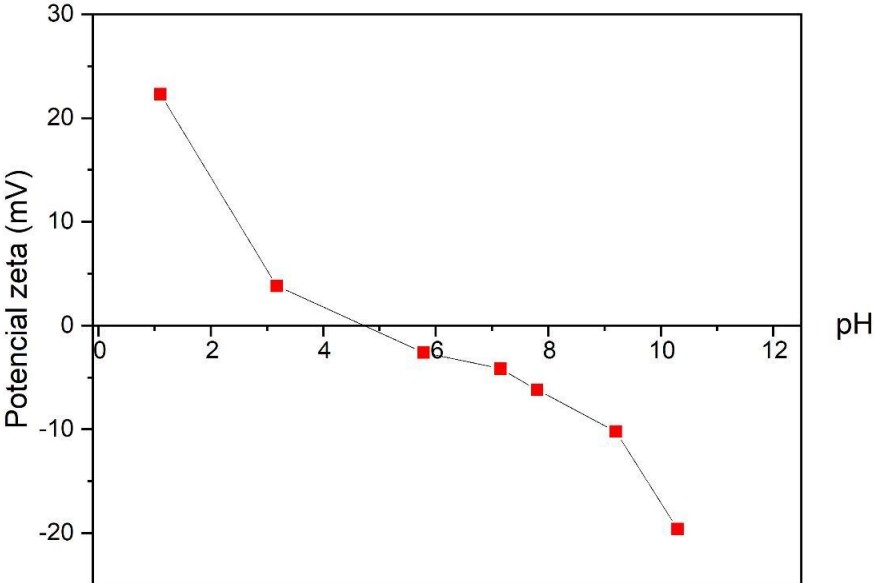

**Figure 6.** Zeta potential of the $CoFe_2O_4$ powders synthesized by the combustion method.

### 3.2. Remazol RR Dye Degradation

The influence of the $CoFe_2O_4$ dosage and $H_2O_2$ concentration on the degradation capacity of the RR dye by the Fenton process is shown in Figure 7.

The precise determination of the catalyst dosage is necessary to evaluate the economics and performance of the material application at the pilot scale. The concentration range from 0 to 4 g/L was selected based on previous studies to determine the ideal catalyst concentration for the present study. The results shown in Figure 7a indicate a significant increase in the efficiency of the catalytic degradation of the RR dye as the catalyst dose was increased up to a certain threshold. The presence of the catalyst was fundamental in the dye degradation process, because the degradation efficiency without the presence of $CoFe_2O_4$ was only 9.9%, whereas that with 0.2 g/L of $CoFe_2O_4$ reached 65.7%. By increasing the amount of $CoFe_2O_4$ from 0.2 g/L to 1.0 g/L, an increase in the dye degradation efficiency from 65.7% to 80.6% was observed. This increase can be attributed to an increase in the number of active sites on the surface of the solid catalyst, which, as expected, accelerated the decomposition of the NPs into oxidizing radicals. In addition, an increase in the catalyst dosage resulted in a greater availability of iron, which can also potentiate the generation of hydroxyl radicals in the Fenton-like reaction [15,54,55].

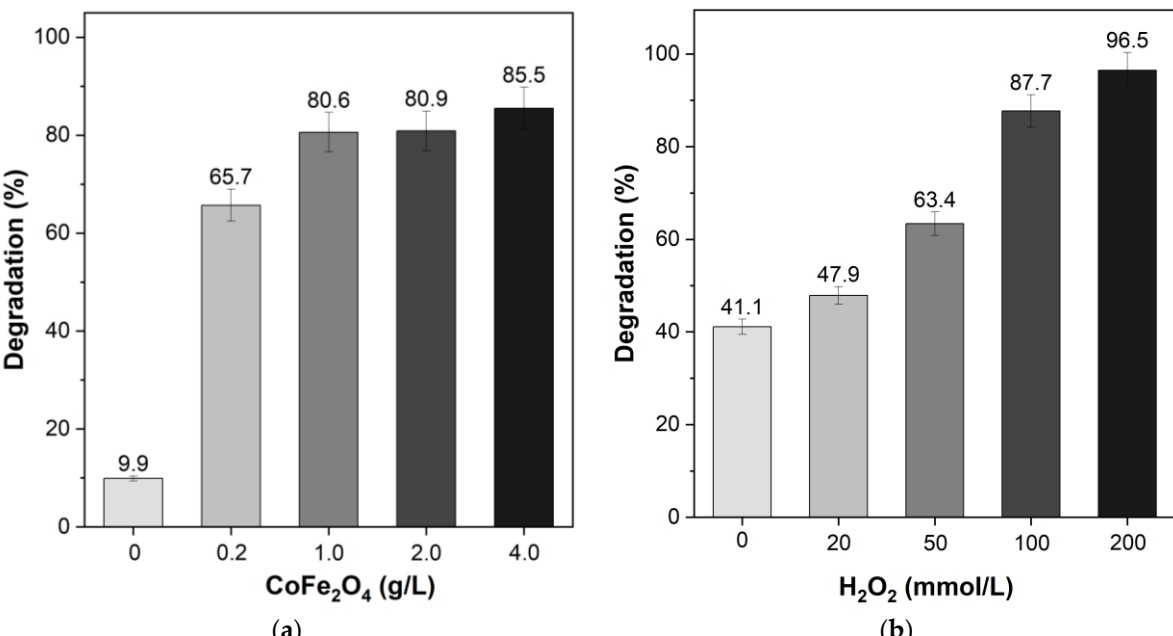

**Figure 7.** RR dye degradation as a function of CoFe$_2$O$_4$ dosages (**a**) and H$_2$O$_2$ concentration (**b**): [RR dye] = 40 mg/L, pH$_{initial}$ = 7.2, t = 24 h and T = 22 ± 1 °C.

However, when the amount of catalyst was increased to 2.0 g/L, the degradation rate remained constant at 80.9%. When the amount was doubled, the maximum degradation rate reached a value of 85.5%, indicating that catalyst doses above 1.0 g/L were not advantageous in terms of process efficiency. Additional amounts of CoFe$_2$O$_4$ above the optimal concentration can lead to the agglomeration of particles in the solution, which reduces the surface area available for radical generation and increases the length of the diffusion path [37]. In addition, increasing the dosage of NPs increased the concentration of available iron in the reaction medium, which may result in undesirable ROS elimination reactions owing to excess iron ions [15,18,55]. Therefore, the optimal CoFe$_2$O$_4$ dosage of 1 g/L was used for further catalytic experiments.

These tests also highlight the remarkable adsorption potential of CoFe$_2$O$_4$. Without the addition of H$_2$O$_2$ (Figure 7b), the removal reached 41.1% with an adsorption capacity of q = 30.2 mg/g. In addition, the presence of H$_2$O$_2$ enhanced the dye removal, suggesting a synergistic effect between Fenton degradation and dye adsorption. Regarding the H$_2$O$_2$ concentration, the RR dye degradation efficiency increased with increasing the H$_2$O$_2$ concentration, reaching a peak of 96.5% at a concentration of 200 mM. H$_2$O$_2$ acts in the system as a reactive oxygen species (•OH and •OOH), as shown in Equations (1)–(3), which describe the dye degradation [56]:

$$Fe^{2+} + H_2O_2 \ \rightarrow \ Fe^{3+} + \bullet OH + OH^- \tag{1}$$

$$Fe^{3+} + H_2O_2 \ \rightarrow \ Fe^{2+} + \bullet OOH + H^+ \tag{2}$$

$$Fe^{2+} + H_2O_2 + \bullet OOH \ \rightarrow \ Fe^{3+} + \bullet OH + OH^- \tag{3}$$

However, a higher-than-ideal dosage of H$_2$O$_2$ can act as a scavenger of the produced hydroxyl radicals (Equations (4)–(6)), resulting in a terminal effect on the dye degradation [57,58].

$$H_2O_2 \ \rightarrow \ \bullet OH + OH^- \tag{4}$$

$$\bullet OH + OH^- \rightarrow H_2O + \frac{1}{2}O_2 \tag{5}$$

$$OH^- + H^+ \rightarrow H_2O \tag{6}$$

Because 200 mM of $H_2O_2$ reached values very close to the maximum degradation capacity, higher values were not tested. Moreover, 100 mM of $H_2O_2$ also showed a significant result, and the difference between 200 mM and 100 mM was less than 10%.

The $Fe/H_2O_2$ ratio is very important for the Fenton reaction [10,59]. In the case of homogeneous Fenton reactions, the typical ranges given in the literature indicate that one part of Fe is required for five to twenty-five parts of $H_2O_2$ (by weight); however, the $Fe^{2+}$ or $Fe^{3+}$ form is irrelevant if there is sufficient $H_2O_2$ and organic material. However, if reduced $H_2O_2$ amounts are used (between 10 and 25 mg/L of $H_2O_2$ or 0.3 and 0.7 mM $H_2O_2$), there may be a preference for the $Fe^{2+}$ ion [60]. For heterogeneous reactions, there is no consensus in the literature on the optimal level, because it depends on the nature of the substance to be degraded and the catalyst.

Considering that 1 g/L of $CoFe_2O_4$ was used as the catalyst, the $Fe:H_2O_2$ weight ratios applied in this study were 1:1.4, 1:3.6, 1:7, and 1:14, i.e., the best results were obtained with respect to the optimal range of conditions presented in the literature, indicating that the Fe available in the catalyst was easily accessible to $H_2O_2$ molecules for radical generation. Based on these results, 200 mM was used for subsequent tests.

Figure 8 shows the kinetic degradation profile of the RR dye and the curves obtained by UV-Vis spectroscopy at different reaction times.

$CoFe_2O_4$ showed adsorptive potential for the RR dye (Figure 8a). Without the addition of $H_2O_2$, the dye concentration was reduced by 35.3%, corresponding to a material adsorption capacity of 16.02 mg/g. The high surface area and pore volume of ferrite available for the adsorption of the RR dye in solution, as well as the surface charge of the material, enhanced the adsorption. This property is desirable in heterogeneous Fenton process catalysts because it helps to concentrate the reagents involved in the Fenton reaction on the material surface, allowing for a more efficient and controlled reaction that is essential for the effective degradation of persistent organic pollutants [61–63].

During the first 10 min after the $H_2O_2$ addition, no decrease in the dye concentration in the solution was observed. Other studies have reported that the addition of $H_2O_2$ to the system induces rapid dye desorption because of competitive adsorption at the binding sites of the catalyst [64–66]. In this study, this increase in the dye concentration was not observed, but the stability of its concentration was observed. It is possible that this behavior was the result of the synchronized effect of dye desorption and catalytic decomposition, in addition to the fact that $H_2O_2$ was not in excess in the system [60]. Between 40 and 120 min, a gradual decrease in the RR dye concentration was observed, followed by a period of less-pronounced degradation up to 270 min, when 75.3% of the dye degradation was reached. This behavior was also observed in the absorbance curves shown in Figure 8b; the characteristic peak at 525 nm and a decrease in the absorbance intensity were identified.

The kinetic data were fitted using the kinetic model proposed by Behnajady et al. [67], as shown in Equation (7) and its corresponding linear form in Equation (8):

$$\frac{C}{C_0} = 1 - \frac{t}{m + bt} \tag{7}$$

$$\frac{t}{1 - \frac{C}{C_0}} = m + bt \tag{8}$$

In this model, C is the RR dye concentration at time t (min), $C_0$ is the initial RR dye concentration, *m* is a dimensionless constant associated with the initial removal rate, and *b* is a dimensionless constant associated with the maximum oxidation capacity.

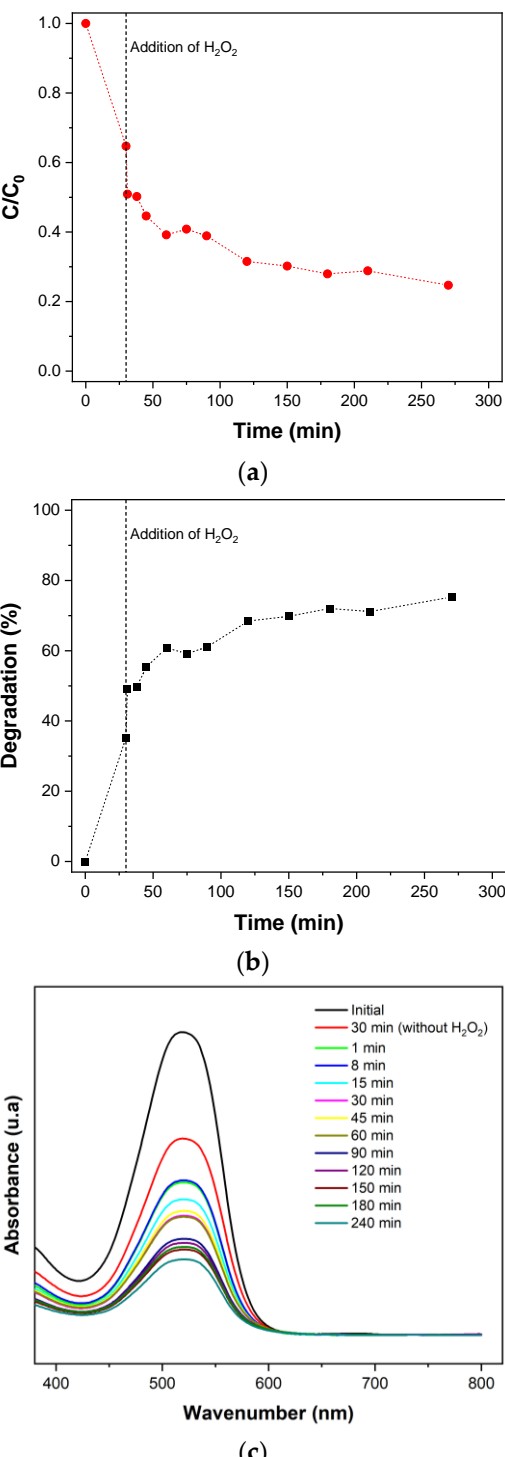

**Figure 8.** (**a**) Degradation kinetics of RR dye by Fenton-like reaction: $[CoFe_2O_4]$ = 1.0 g/L, $[H_2O_2]$ = 200 mM, [RR dye] = 40 mg/L, $pH_{initial}$ = 7.2, and $T$ = 22 $\pm$ 1 °C; (**b**) UV–Vis spectroscopy of RR dye degradation for different times; (**c**) absorbance curve as a function of wavenumber. NOTE: the dotted line indicates the moment of addition of $H_2O_2$.

The results of the kinetic fit to the experimental data, shown in Table 3 and Figure 9, indicate a linear relationship with an excellent model fit to the experimental data ($R^2$ = 0.998). From this curve, the constants m and b of the model could be calculated. The value of 1/m was 0.036, indicating a slower initial decay rate of the RR dye, whereas the maximum

oxidation capacity obtained (1/b) was 0.806. The model also indicated that 1/b and 1/m were inversely proportional to the $Fe/H_2O_2$ ratio.

**Table 3.** Parameters of RR dye degradation using the Behnajady kinetic model [67].

| Parameters | Values |
| --- | --- |
| m | $28 \pm 3$ |
| b | $1.24 \pm 0.02$ |
| $S^2R$ | 190.93 |
| $R^2$ | 0.998 |

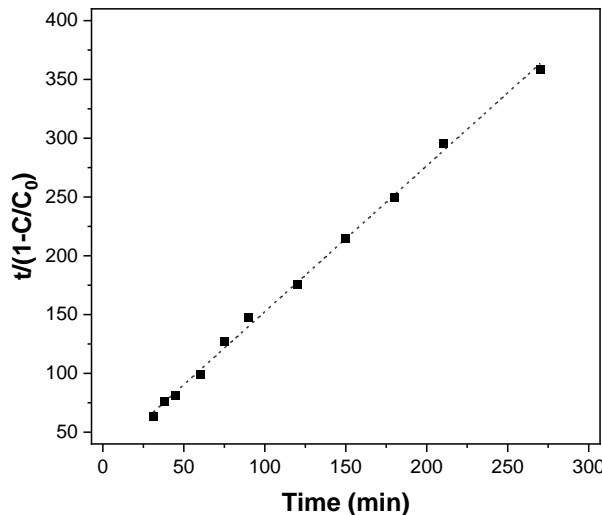

**Figure 9.** Linear fit of RR dye degradation using the Behnajady kinetic model [67].

### 3.3. Catalyst Reusability

The stability and reusability of $CoFe_2O_4$ as a catalyst for the Fenton process were evaluated over three cycles (Figure 10). In the first cycle, the RR dye concentration decreased by 91.14%; in the second and third cycles, the concentration decreased by 87.58% and 62.73%, respectively. The decrease in catalyst activity may have occurred because of possible poisoning of the active sites of the catalyst by the adsorption of organic species, as well as small losses during the separation steps [68]. After the third cycle, the mass loss of the catalyst was $5.5 \pm 0.9\%$, indicating a minimal loss of material between cycles. Importantly, owing to the properties of ferrimagnetic catalysts, it is difficult to release metal ions into an aqueous solution, which confirms the sustainability of the catalyst. Other studies showing similar degradation behaviors indicate that although a partial loss of degradation efficiency was observed after the third cycle, this efficiency can be increased by extending the reaction time [69,70].

Although the parameters need to be optimized to make the reuse cycles more effective in degrading the RR dye after the third cycle, the results show that the cobalt ferrites used in this study are stable heterogeneous Fenton catalysts capable of magnetic separation and have the potential for long-term application in wastewater treatment.

Typically, nanoparticles have been shown to have toxic potential in aquatic organisms, such as zebrafish and mini-crustaceans [71]. However, $CoFe_2O_4$ nanoparticles, which are magnetic and recoverable, have a lower rate of release into the environment, thus minimizing the environmental impact. The mass of the nanoparticles remained almost constant (with a loss of approximately 5%) after several reuse cycles. Although the catalytic efficiency decreased, the ferrimagnetism was not significantly affected, and the particles continued to be separated by the magnet. The observed loss of efficiency was probably due to the deactivation of the photocatalytic sites. Furthermore, recoverability is an important

advantage of our material because of its magnetic nature, unlike other photocatalysts such as $TiO_2$, which have demonstrated toxicity [72] and are not recoverable in nanometric form.

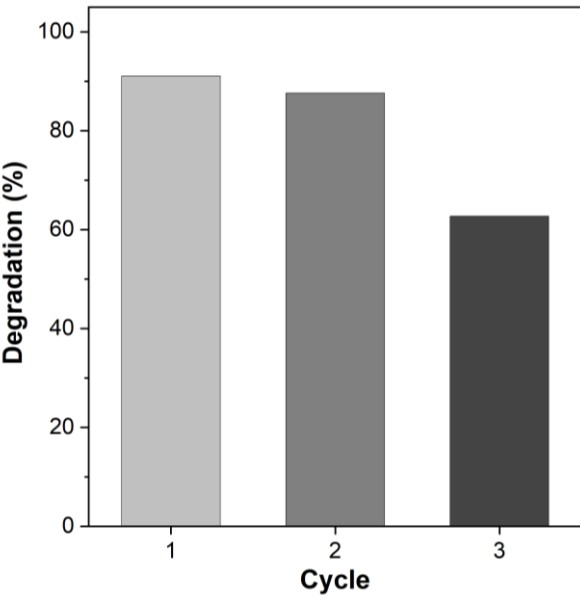

**Figure 10.** Degradation of RR dye by cycle: $[CoFe_2O_4]$ = 1.0 g/L, $[H_2O_2]$ = 200 mM, $t$ = 270 min; [RR dye] = 40 mg/L, pHinitial = 7.2, and $T$ = 22 $\pm$ 1 °C.

### 3.4. RR Dye Degradation Pathway

Assessing the reaction intermediates of RR dyes is valuable for evaluating the effectiveness of catalytic systems and offers insights into the degradation process. The analyses were performed using mass spectrometry (ESI$^-$), and the time-dependent degradation was determined by identifying the peaks formed in the spectra of the solutions incubated for 1–240 min, along with their respective controls, as shown in Figure 11.

Each degradation product had different kinetics, with the compounds with the highest molar mass appearing first, followed by the compounds with the lowest molar mass throughout the degradation kinetics. The maximum product-formation time was 45–60 min. Twelve compounds derived from Remazol dye were identified, and a possible route for the degradation process was proposed.

Initially, the cleavage of the azo bond was observed, leading to the formation of two aromatic compounds, 2-(benzenesulfonyl)ethyl sulfate ($m/z$ 265) and 3-{2-[(hydroxysulfanyl)oxy]ethanesulfonyl}aniline ($m/z$ 248). Waghmode et al. [73] reported 2[(3-aminophenyl) sulfonyl] ethane sulfonic acid in the range of $m/z$ of 265. The final compound underwent deamination and dehydroxylation to form 2-(benzenesulfonyl)ethane- 1-OS-thioperoxol ($m/z$ 218) or desulfurization to generate 2-(3-aminobenzene-1-sulfonyl)ethane-1-ol ($m/z$ 201) or (3-aminobenzene-1-sulfonyl)acetaldehyde ($m/z$ 199). There was a time-dependent formation of naphthalene-1-sulfinate ($m/z$ 191), 2-aminonaphthalene-1-ol ($m/z$ 158), naphthalene-1-ol ($m/z$ 143), 1,2-dihydronaphthalene-1-ol ($m/z$ 145), tetrahydronaphthalene-1-ol ($m/z$ 147), 3-methylidenepent-4-en-2-ol ($m/z$ 97), and butane-2-thiol ($m/z$ 89). This final degradation behavior, in which the structures of phenolic compounds prevailed, shows that the results were in accordance with previous studies. Different authors have reported that in smaller structures, asymmetric cleavage occurs, forming 2-amino naphthalene ($m/z$ 141), N-phenyl-1, 3, 5 triazine ($m/z$ 170), and aniline ($m/z$ 93) [73–77]. The presence of these simple organic compounds shows that the Fenton process was not interrupted after the decomposition of the organic dye but proceeded towards the complete mineralization of organic matter to form $CO_2$ and $H_2O$.

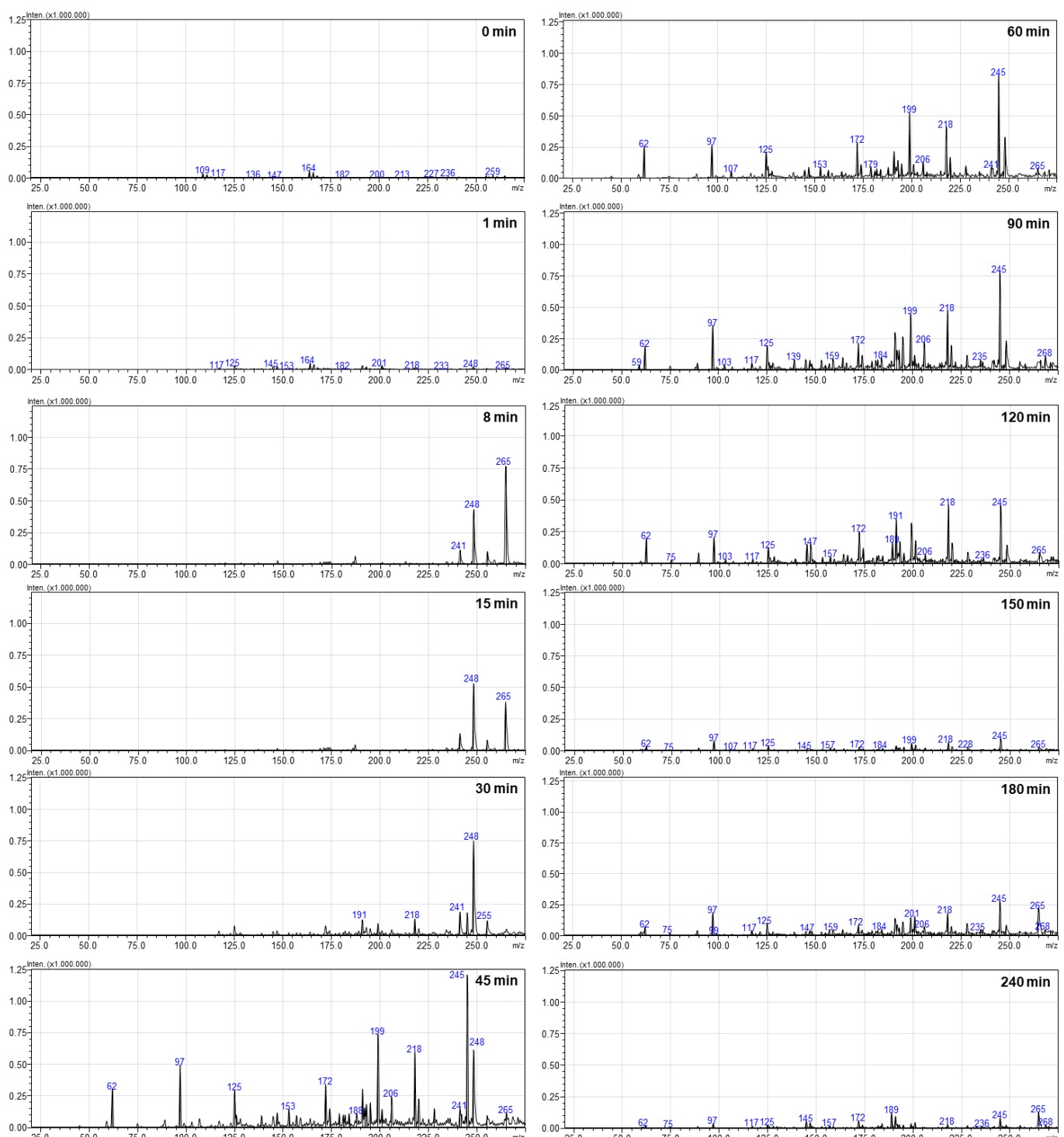

**Figure 11.** Mass spectra (full scan) of the time-dependent degradation of RR dye obtained by tandem mass spectrometry.

These results demonstrate the outstanding capability of $CoFe_2O_4$ for dye degradation in a simple water-based solution. This groundbreaking study presents the first report on RR dye degradation using $CoFe_2O_4$ as a catalyst in the Fenton process. Additionally, no studies have used $CoFe_2O_4$ to degrade dyes in real wastewater treatment scenarios. As highlighted above, basic degradation products can be produced via different degradation pathways. This is also expected when using real wastewater containing other substances in the solution

that reduce the degradation efficiency of the compound. Other studies have identified this effect using several catalysts [9]. The presence of inorganic ions, suspended solids, and organic matter can affect the selectivity of $CoFe_2O_4$ towards the dye. In addition, the effectiveness of the material as a catalyst can be affected by various operating conditions such as temperature, pH, and oxygen concentration. Thus, ferrite has been proposed as a Fenton catalyst for effluent polishing (tertiary treatment). To meet legally mandated effluent emission requirements for bodies of water or for potential reuse in industrial processes, it should be used after conventional treatment. Therefore, it is recommended to use this degradation process as an effluent-polishing process (tertiary treatment) after conventional treatment to achieve the conditions required by law for the discharge of the effluent into a body of water or even its reuse in industrial processes.

## 4. Conclusions

The results presented in this study demonstrate the promising application of $CoFe_2O_4$ nanoparticles synthesized using the combustion method as effective catalysts in the Fenton process for the degradation of persistent textile dyes in contaminated water.

The synthesized $CoFe_2O_4$ nanoparticles exhibited well-defined crystalline structures, high surface areas, and ferrimagnetic properties, which are essential for their catalytic performance in the Fenton process. The $CoFe_2O_4$ showed a remarkable adsorption capacity for the RR dye, and the addition of $H_2O_2$ resulted in significant dye degradation rates within 240 min. The Behnajady kinetic model was successfully used to describe the degradation behavior of the dye over time. The material was reusable over multiple reaction cycles, although its degradation efficiency decreased after the third cycle. However, the results suggest that, with parameter optimization, it is possible to improve the efficiency in subsequent cycles.

The degradation products of the RR dye were analyzed by chromatographic methods, showing that the degradation of the organic compound by the Fenton process proceeded towards complete mineralization. In summary, the efficacy of $CoFe_2O_4$, its potential for reuse, and its favorable physicochemical properties pave the way for its practical application in wastewater treatment, contributing to a reduction in the environmental impacts caused by the textile industry and other sources of organic pollution. In addition, the optical properties of the catalyst may open the door for its future applications in advanced photocatalytic processes.

**Author Contributions:** Conceptualization, M.A.P.C., E.A.C.M., S.E. and T.B.W.; Methodology, J.L.N., P.M.T., T.f.d.O. and F.R.-P.; Validation, M.A.P.C., P.M.T., E.A.C.M. and O.R.K.M.; Formal analysis, M.A.P.C., P.M.T., E.A.C.M., S.E., O.R.K.M. and T.B.W.; Investigation, M.A.P.C. and J.L.N.; Resources, F.R.-P. and S.A.; Data curation, T.B.W.; Writing—original draft, M.A.P.C., J.L.N. and T.B.W.; Writing—review & editing, S.E., T.F.D.O., F.R.-P., O.R.K.M., T.B.W. and S.A.; Supervision, E.A.C.M., T.F.D.O., F.R.-P., O.R.K.M. and S.A.; Project administration, S.A.; Funding acquisition, S.A. All authors have read and agreed to the published version of the manuscript.

**Funding:** The authors are very grateful to Universidade do Extremo Sul Catarinense (UNESC), Coordination for the Improvement of Higher Education Personnel (Coordenação de Aperfeiçoamento de Pessoal de Nível Superior, CAPES/Brazil), National Council of Technological and Scientific Development (Conselho Nacional de Desenvolvimento Científico e Tecnológico, CNPq/Brazil; processes n. 161197/2020-5, 307702/2022-7, and 310328/2020-9), Laboratório Integrado de Meio Ambiente (LIMA/UFSC) and FAPESC (T.O. 2021TR1650, T.O. 2021TR001314, T.O 2021TR001817) for supporting this work.

**Data Availability Statement:** Not applicable.

**Conflicts of Interest:** The authors declare no conflict of interest.

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
