# Peer review of "Cobalt Ferrite (CoFe2O4) Spinel as a New Efficient Magnetic Heterogeneous Fenton-like Catalyst for Wastewater Treatment"

_sustainability, doi:10.3390/su152015183_

Round 1

Reviewer 1 Report

The manuscript titled "Cobalt ferrite (CoFe2O4) spinel as a new efficient magnetic heterogeneous Fenton-like catalyst for wastewater treatment" presents an intriguing study on the synthesis and application of CoFe2O4 nanoparticles as a catalyst for Red Remazol RR dye degradation using the Fenton process. Overall, the study is well-conducted and provides valuable insights into the potential applications of the synthesized material. However, there are a few minor revisions that should be addressed before publication.

1.      The introduction provides a clear context for the study. However, it would be beneficial to briefly highlight the significance of using CoFe2O4 as a catalyst for wastewater treatment, especially in comparison to existing catalysts. This would help the reader better understand the motivation behind the research.

2.      It is necessary to adjust Figure 1. The current form of Figure 1 seems to have a lot of gaps. I suggest that the author can adjust the display of the structural formula and add a 3D model of the molecule next to the structural formula.

3.      For the discussion of Ψ-dependent magnetic properties, consider providing more explanation of the observed trends and their relevance to the catalytic application.

4.      Mention more potential challenges or limitations associated with using CoFe2O4 as a catalyst in real wastewater treatment scenarios.

5.      The Co-based metal oxide catalysts are also reported by different groups, such as Adv. Mater. (10.1002/adma.202305074); Adv. Funct. Mater. (10.1002/adfm.202207618); J. Energy Chem. (10.1016/j.jechem.2023.03.033); Appl. Phys. Rev. (10.1063/5.0083059), which could be mentioned in the introductions.

Minor editing of English language required

Author Response

Enclosed you can find our revised manuscript for your consideration and possible publication in Sustentability. Please note that we have addressed the ideas suggested by the reviewers. We strongly appreciate these suggestions and comments, which we find to be very constructive and helpful and have contributed to strengthen our manuscript. Based on these suggestions and comments, we have made the corresponding modifications in the paper, which were highlighted in in gray for reviewer 1, blue for reviewer 2 and green for reviewer 3. The detailed response to the reviewers’ comments is given in the following.

Reviewer 2 Report

Thank you for giving me the opportunity to revise the MS entitled “Cobalt ferrite (CoFe2O4) spinel as a new efficient magnetic heterogeneous Fenton-like catalyst for wastewater treatment” by Maria and his/her colleagues that was submitted to “Sustainability”.  The MS submitted is suitable for Sustainability, and some interesting results were showed. However, there are several suggestions that could be considered by the authors.

1.         I think the introduction section should be revised better. the innovation of the manuscript must be clearly stated in the text.

2.         Line 344-354 Please check the format of H2O2.

3.         The manuscript should discuss whether separate H2O2 and CoFe2O4 have removal effects on RR dye.

4.         Figure 6 should indicate the reaction conditions.

5.         Figure 7 a Just keep one for C/C0 and Degradation.

6.         RR dye degradation pathway requires more in-depth comparison and discussion.

7.         Suggest carefully checking the format and language of the entire text.

8.         The format of references should be unified, such as capitalization, superscribing, and page numbers.

9.         I would suggest that the authors review and include the following recent studies to improve the manuscript. (Water Science & Technology, 2018, 77 (9), 2174-2183, Nanomaterials. 2020, 10(9), 1719, International Journal of Environmental Research and Public Health 2022, 19(19), 12354).

Best regards,

Author Response

(The authors gave the same response as above.)

Reviewer 3 Report

The current work focuses on Cobalt Ferrite (CoFe2O4) Spinel as a New Efficient Magnetic Heterogeneous Fenton-like Catalyst for Wastewater Treatment. The author’s great effort into the manuscript, but minor issues should be addressed. In addition, extensive editing of the English language is required

Abstract

The use of nano-spinel ferrites for water treatment applications has been extensively studied. First, show the current manuscript's novelty and importance and then the main outputs.

-Line 18, the first appearance of the abbreviation should have a full definition e.g. CoFe2O4

Keywords

- “Cobalt Ferrite” should be inserted 

Introduction 

-The introduction doesn’t provide sufficient background and all relevant references are not included.

https://doi.org/10.3390/su15054586

https://doi.org/10.1016/j.heliyon.2022.e09654

-The novelty of this work is not highlighted and the author's contribution was unclear compared to other previous works. 

Materials and Methods

-What is the scan rate for XRD analysis?

Results and discussion

-TEM should inserted to investigate the morphology of the prepared nanoparticles.

-What about the toxicity of the fabricated nanoparticles, which is an important factor for materials used in water treatment?

-Indexed peak position should inserted in Fig 2a

-Insert error bar for Fig. 6

- page 11, correct typo H2O2

-The efficiency of the fabricated nanoparticles should be compared to other previous works.

Extensive editing of the English language is required

Author Response

(The authors gave the same response as above.)

Round 2

Reviewer 2 Report

ok to accept. Please check the format of the entire text, such as superscripts and subscripts.